# Revision of Frequency Estimates of Extreme Precipitation Based on the Annual Maximum Series in the Jiangsu Province in China

**Yuehong Shao [1,\*], Jun Zhao [1,2], Jinchao Xu [1,2], Aolin Fu [1] and Junmei Wu [3]**

[1] School of Hydrology and Water Resources, Nanjing University of Information Science and Technology (NUIST), Nanjing 210044, China; zsmzyq@126.com (J.Z.); xujinchao301@foxmail.com (J.X.); -fual177@126.com (A.F.)

[2] Nanjing Hydraulic Research Institute, Nanjing 210029, China

[3] Kunshan Meteorology Bereaus, Kunshan 215300, China; wjmqingkong@126.com

\* Correspondence: syh@nuist.edu.cn; Tel.: +86-138-1589-7365

**Abstract:** Frequency estimates of extreme precipitation are revised using a regional L-moments method based on the annual maximum series and Chow's equation at lower return periods for the Jiangsu area in China. First, the study area is divided into five homogeneous regions, and the optimum distribution for each region is determined by an integrative assessment. Second, underestimation of quantiles and the applicability of Chow's equation are verified. The results show that quantiles are underestimated based on the annual maximum series, and that Chow's formula is applicable for the study area. Next, two methods are used to correct the underestimation of frequency estimation. A set of rational and reliable frequency estimations is obtained using the regional L-moments method and the two revised methods, which can indirectly provide a robust basis for flood control and water resource management. This study extends previous works by verifying underestimation of the quantiles and the provision of two improved methods for obtaining reliable quantile estimations of extreme precipitation at lower recurrence intervals, especially in solving reliable estimates for a 1-year return period from the integral lower limit of the frequency distribution.

**Keywords:** regional l-moments; revision of frequency estimation of extreme precipitation; chow's equation; annual maximum series; annual exceedance series





## 1. Introduction

Natural flood disasters occur frequently in China. As a consequence, flood control is an important topic relevant to the preservation of human life, property, and society [1,2]. Scientific and robust flood control standards are critical to engineering and urban flood control design, for which an important theoretical basis of estimation is hydrological frequency calculation [3]. Rapid economic development and enhanced environmental consciousness have led to increased attention on extreme hydrometeorological events and growing concern for events occurring at lower return periods, fueled by the increasing seriousness of urban waterlogging disasters [1,4]. However, the sampling method and the choice of probability distribution can influence frequency estimations at low recurrence intervals [5]. Therefore, knowledge regarding sampling and the optimum distribution is a key element of frequency analysis.

The determination of an appropriate distribution is an important step in frequency analysis. Selection of the optimum distribution has been extensively researched because the theoretical distribution curve is unknown [6,7]. For example, the person type III (PE3) distribution has been selected as the appropriate fit in the United States, and the generalized extreme value (GEV) distribution has been recommended in more than ten countries [8,9]. However, the adoption of a "one-size-fits-all" scenario may lead to poor accuracy in quantile estimation due to heterogeneity and discordancy associated with different sites

in the region [10]. Therefore, some researchers have recommended selecting an optimum distribution for each homogeneous region based on practical grounds, showing that the accuracy of quantile estimates is significantly improved [8,11,12]. In this study, Monte Carlo (MC) simulations and a diagram of L-moments ratios are used to determine goodness-of-fit. However, the criterion from MC simulation can be unreliable when data are serially linked or cross-correlated among sites [13]. Therefore, a summary assessment is performed using different statistical criteria to determine the optimum distribution, in order to avoid obtaining an arbitrary result from any single test.

For sampling, the annual maximum series (AMS) and partial duration series (PDS) can be used to select extreme values from a long hydrological time series. In the AMS, the largest event in each year is extracted and recorded in a series that contains critical information such as extreme precipitation or peak flow amount. These data are easily obtained and widely used in hydrological statistical analysis [14–17]. However, the AMS extracts only the largest event, and secondary events occurring in one year may exceed the annual maximum of other years. In addition, annual maximum events observed in dry years may be very small, and interpretations based on these events can lead to significant bias with respect to the outcome of an extreme value analysis [18–20]. Extensive research has shown that quantiles based on AMS data are underestimated to a certain extent at low return periods [1,12,21]. For example, Lin et al. showed underestimation of extreme precipitation based on AMS data from daily precipitation data obtained from 1438 stations in the southwestern United States [20]. However, less research has been done to assess the underestimation of quantiles in China, or to verify the results in a specific area [22].

The PDS method extracts all of the extreme events above a truncation or threshold level for the analysis and therefore does not suffer from the drawbacks inherent to the AMS data. If a descending sort of PDS is selected, such that the number of values in the series equals the number of years on record, the series is called the annual exceedance series (AES). The AES data not only simplifies sample selection and subsequent statistical analysis, but also gives similar results to that obtained with PDS data and can be regarded as a special case of the PDS [20]. A complete description and solid theoretical basis of precipitation and flood processes exists in the PDS. Previous research has shown that the PDS is more efficient for quantile estimation than the AMS because it is more suitable for a heavy-tailed distribution, which is common in hydrological applications [23–25]. The accuracy of estimation based on the PDS is closely related to the selection of an appropriate threshold level and independence of the sample data [26–28]. Construction of a PDS model can be hampered by several difficulties and is less commonly used in hydrologic research than AMS methods. First, events should be independent; hence, criteria explicitly identifying independent events must be defined. Second, the selection of an appropriate threshold is important to the result and should ensure that a maximum amount of relevant information is included in the analysis without violating basic statistical assumptions. Third, the return period of the PDS in sampling units is not consistent with the return period of the AMS in years. Conversion and verification of recurrence intervals are difficult with PDS data [19,29,30].

Although it has a solid theoretical base, difficulties such as data availability make the construction of a PDS model difficult. At most sites in China, only AMS data are available for a variety of reasons. Therefore, the development of a simple and feasible method for calculating reliable quantile estimates on the basis of AMS data is critical. Chow [21] derived a relation for AMS and AES between two recurrence intervals corresponding to the same event that has been widely accepted and used in engineering practice [31]. Lin et al. subsequently verified the applicability of Chow's equation in the southwestern United States [20]. Takeuchi noted that the precision of estimations can satisfy the requirement using the return period in sampling units of Chow's equation if the size of the PDS is in accordance with a Poisson distribution [32]. Ghahraman noted that the relationship between the recurrence intervals of AMS and PDS should be a function of rainfall duration and the number of samples [33]. Is the frequency conversion related to local hydrological

characteristics and other factors? Is Chow's formula applicable to hydrological frequency analysis in China? These questions require in-depth analysis and research. If Chow's formula is appropriate for sites in China, calculation of reliable frequency estimates based on the AMS data and Chow's formula can be performed. Because only AMS data exist for most sites in China, this highlights the purpose and importance of this study.

The objectives of this study are to verify the underestimation of quantiles, provide two revised methods for reliably estimating the frequency of extreme precipitation, and to solve the problem of the distribution integral lower limit. To achieve these objectives, different homogeneous regions are first identified and the optimum distribution for each homogeneous region in the study area is determined. Second, we compare exceedance frequencies with the exceedance probabilities in order to verify whether quantiles are underestimated based on AMS data in the study area. Third, frequency estimates are computed using real AMS data, real AES data, and generated AES data based on the Chow's equation and AMS data, to verify the applicability of Chow's equation in this study. Last, a set of reliable frequency estimates is obtained using a regional L-moments method based on AMS data and Chow's equation. We revise the estimation of quantiles at each site at low return periods for the AMS data and also solve the quantiles for a 1-year return period; the latter is a major merit of this research that extends previous work conducted to date.

## 2. Materials and Methods

### 2.1. Study Area

An important part of Yangtze River Delta, Jiangsu Province (116°18′–121°57′ E, 30°45′–35°20′ N) covers an area of about $1.07 \times 105$ km$^2$ and is located downstream of Yangtze River and Huaihe River basins. The terrain is dominated by plains, accounting for more than 70% of the area. Hills are concentrated in the southwest, accounting for 14.3% of the total area. The terrain slopes from west to east. The river network is intricate and includes the three major river systems of the Yishusi River drainage: Downstream of the Huaihe River, the Yangtze River, and Taihu Lake stream. Jiangsu is located in a transitional subtropical to warm temperate climate zone. The area is characterized by four distinct seasons, which are cold and dry in winter, and warm and humid with plum rains in the late spring and early summer, and typhoons in summer and autumn. The annual average rainfall is 996 mm. Precipitation gradually increases from south to north and is greater on the coast than inland. Rainstorm zones are mainly located in the south of Yimeng Mountain. The elevation, stream network, and meteorological stations of the study area are shown in Figure 1.

### 2.2. Data

Daily precipitation from meteorological stations was obtained for this study from the National Meteorological Information Centre of the China Meteorological Administration (http://cdc.cma.gov.cn/shuju). Data from 63 representative stations in the Jiangsu area obtained between 1961 and 2011 were used for analysis. The AMS was extracted from the daily precipitation data using a bubble sort method. The annual maximum series $x_1$, $x_2$, $x_3$, . . . . . . , $x_N$ is a collection of the maximum data for each year, where $N$ is the number of years in the observed time series. The partial duration series $y_1$, $y_2$, $y_3$, . . . . . . , $y_M$ is a collection of exceedance over a certain truncation level, the $M^{th}$ largest in the whole time series of $N_{yr}$. In this study, the threshold value was equal to three. That is, the three largest daily rainfalls were selected from each year and form the PDS. The PDS was sorted and intercepted the largest $N$ events in descending order, which includes the AES. Therefore, the AES may be regarded as a special case of the PDS. The frequency estimations for extreme precipitation based on AMS and AES were assessed and compared.

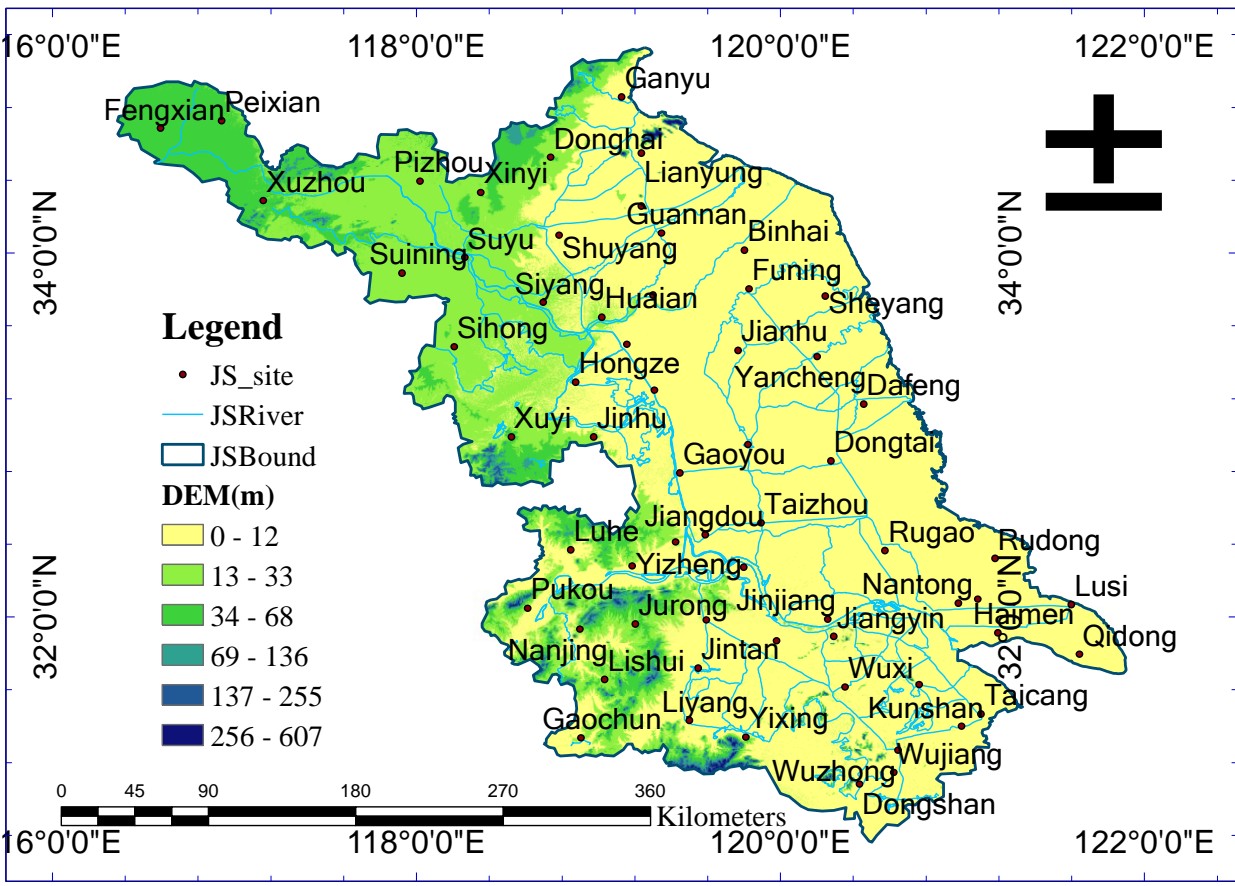

**Figure 1.** Map showing the locations of meteorological stations in Jiangsu province.

*2.3. Methodology*

2.3.1. Regional L-Moments Method

The L-moments method is aimed at the issue of the robust parameter estimation. Regional analysis provides a solution to reduce the uncertainties that exist in at-site statistical analysis. Accordingly, many studies have shown that a regional L-moments method is a reasonable and reliable method to improve the precision and accuracy of frequency estimation [12,20].

Regional frequency analysis employs data from several sites in a region to estimate the frequency distribution of the underlying population at each site. The approach makes the assumption that the shape of the probability distribution function is shared among a group of sites. An index-flood procedure was used in the estimation of precipitation frequency. It assumes that the frequency distribution at each of the $N$ sites in a region is identical apart from a site-specific scaling factor, the index-flood, and that the region is homogeneous. That is, the quantile estimates at site $i$, $Q_{T,j,i}$ can be computed by a regional component that reflects the common precipitation character and a local component that reflects the site-specific scaling factor. The formula can be written as:

$$Q_{T,j,i} = q_{T,j} \times \quad \overline{x}_{i,j} \tag{1}$$

where $\overline{x}_{i,j}$ is commonly the at-site sample mean used for the location estimator, $j = 1, 2, \ldots, N$; $q_{T,j}$, namely the regional growth factor (RGF), is defined as the dimensionless regional frequency distribution common to the N sites in the region at multiple desired return periods, $T_j$. It can be determined by a set of regional parameters that are weighted average values over $N$ sites for a selected distribution. For example, the regional Linear coefficient of deviation ($L\text{-}C_v$) can be written as follows:

$$\hat{L}_{Cv}{}^{(R)} = \sum_{i}^{N} n_i \hat{L}_{Cv}{}^{(i)} / \sum_{i}^{N} n_i, \ i = 1, 2, \ldots, N \tag{2}$$

where $\hat{L}_{Cv}{}^{(R)}$ and $\hat{L}_{Cv}{}^{(i)}$ are respectively denoted as the regional $L$-$C_v$ and the single station $L$-$C_v$ at site $i$.

### 2.3.2. Identification of Homogeneous Regions

The identification of homogeneous regions is an important task. First, cluster analysis is used to identify homogeneous regions on the basis of four variables: Longitude, latitude, elevation, and the mean annual precipitation. This analysis is conducted using Ward's method based on Euclidean distance by Statistical Analysis System (SAS) hierarchical clustering software [34] More details on cluster analysis can be found in reference [13]. Second, a measurement of heterogeneity ($H$) is used to assess hydrological similarity and determine regional homogeneity. H is denoted as:

$$H_i = (V_i - \mu_{V_i}) / \sigma_{V_i} \ i = 1, 2, 3 \tag{3}$$

where $\mu_{V_i}$ and $\sigma_{V_i}$ are the expectation and standard deviation of $V_i$, which can be defined as follows:

$$
\begin{aligned}
V_1 &= \left\{ \textstyle\sum_{i=1}^{N} n_i (t^{(i)} - t^R)^2 / \sum_{i=1}^{N} n_i \right\}^{1/2}, \\
V_2 &= \textstyle\sum_{i=1}^{N} n_i \left\{ (t^{(i)} - t^R)^2 + (t_3^{(i)} - t_3^R)^2 \right\}^{1/2} / \sum_{i=1}^{N} n_i, \\
V_3 &= \textstyle\sum_{i=1}^{N} n_i \left\{ (t_3^{(i)} - t_3^R)^2 + (t_4^{(i)} - t_4^R)^2 \right\}^{1/2} / \sum_{i=1}^{N} n_i.
\end{aligned}
\tag{4}
$$

where $t^{(i)}$, $t_3^{(i)}$, and $t_4^{(i)}$ are separately the coefficient of sample L-moments. $t^R$, $t_3^R$, and $t_4^R$ denoted regional average L-moments coefficient weighted the site's record lengths, which are defined as:

$$t^R = \sum_{i=1}^{N} n_i t^{(i)} / \sum_{i=1}^{N} n_i, \quad t_3^R = \textstyle\sum_{i=1}^{N} n_i t_3^{(i)} / \sum_{i=1}^{N} n_i, \quad t_4^R = \textstyle\sum_{i=1}^{N} n_i t_4^{(i)} / \sum_{i=1}^{N} n_i \tag{5}$$

where $N$ is the number of sites, $n_i$ is the site's record lengths. Hosking and Wallis [13] suggested that a region may be considered "acceptable homogeneous" if $H < 1$, "possibly heterogeneous" if $1 \leq H < 2$, "definitely heterogeneous" if $H \geq 2$, and "possibly correlated" if $H < 0$.

Finally, a measurement of discordancy ($D_i$) is used to identify data that are grossly discordant with the region as a whole [13]. The critical values for discordancy experiments are dependent on the number of sites in the region [13]. More detailed information on these procedures can be found in [12,13].

### 2.3.3. The Goodness-of-Fit

The MC simulation and the Root Mean Square Error ($RMSE$) of the sample L-moments are used to determine the appropriate distribution according to an arbitrary result of any one test. Due to the relative stability and flexibility of 3 parameters, five kinds of commonly used 3-parameter distributions are investigated for goodness-of-fit as follows [12,13]: Generalized logistic (GLO), GEV, generalized normal (GNO), generalized pareto (GPA), and PE3.

A large number of synthetic datasets are generated by MC simulation and used to access the deviation from the mean point to the distribution in $L$-$C_k$ scale. For each distribution, the goodness-of-fit measure is defined as follows:

$$Z^{DIST} = \left( \tau_4^{DIST} - t_4^R + B_4 \right) / \sigma_4 \tag{6}$$

where $t_4^R$ is the regional average L-kurtosis, weighted proportionally to the site's record length; $\tau_4^{DIST}$ is the L-kurtosis of the fitted distribution, where DIST can be any of GLO, GEV, GNO, GPA, and PE3. For the $m^{th}$ simulated region, after the regional average L-kurtosis $t_4^{[m]}$ obtained, the bias ($B_4$) and standard deviation ($\sigma_4$) of $t_4^R$ can be calculated as follows:

$$B_4 = \left[ \sum_{m=1}^{N_{sim}} \left( t_4^{[m]} - t_4^R \right) \right] / N_{sim} \tag{7}$$

$$\sigma_4 = \left\{ \left[ \sum_{m=1}^{N_{sim}} \left( t_4^{[m]} - t_4^R \right)^2 - N_{sim} B_4^2 \right] / (N_{sim} - 1) \right\}^{1/2} \tag{8}$$

Assuming that $Z$ takes the form of a standard Gaussian distribution, the criterion $|Z| \leq 1.64$ is chosen as the cutoff threshold, and the smallest $|Z|$ value, the best distribution.

The *MC* simulation emphasizes the effect of the regional average. The *RMSE* is used to assess the variability of the sample $L$-$C_k$ of the real data at $N$ sites to accurately evaluate the distribution pattern. The *RMSE* is calculated for each of the plausible distributions as follows:

$$RMSE = \left\{ \sum_{i=1}^{N} n_i (S_{i,L-Ck} - D_{i,L-Ck})^2 / \sum_{i=1}^{N} n_i \right\}^{1/2} \tag{9}$$

where $S_{i,L-Ck}$ is the sample $L$-$C_k$ at site $i$ and $D_{i,L-Ck}$ is the distribution's $L$-$C_k$ at sample $L$-$Cs$ of site $i$. The distribution with the smallest *RMSE* is selected as the most appropriate distribution based on this experiment. More details of the *MC* and *RMSE* methods can be found in the literature [12,13,20].

### 2.3.4. Conversion of AES-AMS

Chow [21] derived a relation between the two recurrence intervals $T_{AMS}$ and $T_{AES}$ corresponding to the same event, as follows:

$$T_{AES} = \left[ \ln \left( \frac{T_{AMS}}{T_{AMS} - 1} \right) \right]^{-1} \text{ or } T_{AMS} = \frac{1}{1 - e^{-\frac{1}{T_{AES}}}} \tag{10}$$

where $T_{AMS}$ and $T_{AES}$ are, respectively, the return period of AMS and AES.

Chow's equation is a frequency conversion relation that has been widely adopted for use in engineering research. Table 1 gives the return periods based on AES data. Frequency estimation can be computed by using non-exceedance probability ($P_{NON}$). However, the computer program cannot be computed if $P_{NON}$ equals zero. From Equation (10), it is clear that it is not computable for a 1-year event under AMS data. If Chow's equation is applicable to this study area, we can not only correct the frequency estimation at low return periods based on AMS data, but can also compute quantiles for a 1-year recurrence interval based on Chow's equation.

**Table 1.** Return periods based on AMS data.

| $T_{AES}$ (−year) | $T_{AMS}$ (−year) | P = 1/$T_{AMS}$ | $P_{NON}$ = 1−1/$T_{AMS}$ |
|---|---|---|---|
| N/A | 1 | 1.0 | 0.0 * |
| 1.44 | 2 | 0.50 | 0.50 |
| 4.48 | 5 | 0.20 | 0.80 |
| 9.49 | 10 | 0.10 | 0.90 |
| 24.50 | 25 | 0.04 | 0.96 |
| 49.50 | 50 | 0.02 | 0.98 |

* Note: It is incomputable for 1-year event based on AMS data.

## 3. Results

### 3.1. Results and Analysis of the Goodness-of-Fit

The study area was divided into five homogeneous regions according to the above-mentioned procedures in methods section (Figure 2). After identification of homogeneous regions, the optimum distribution is determined based on the regional L-moments analysis.

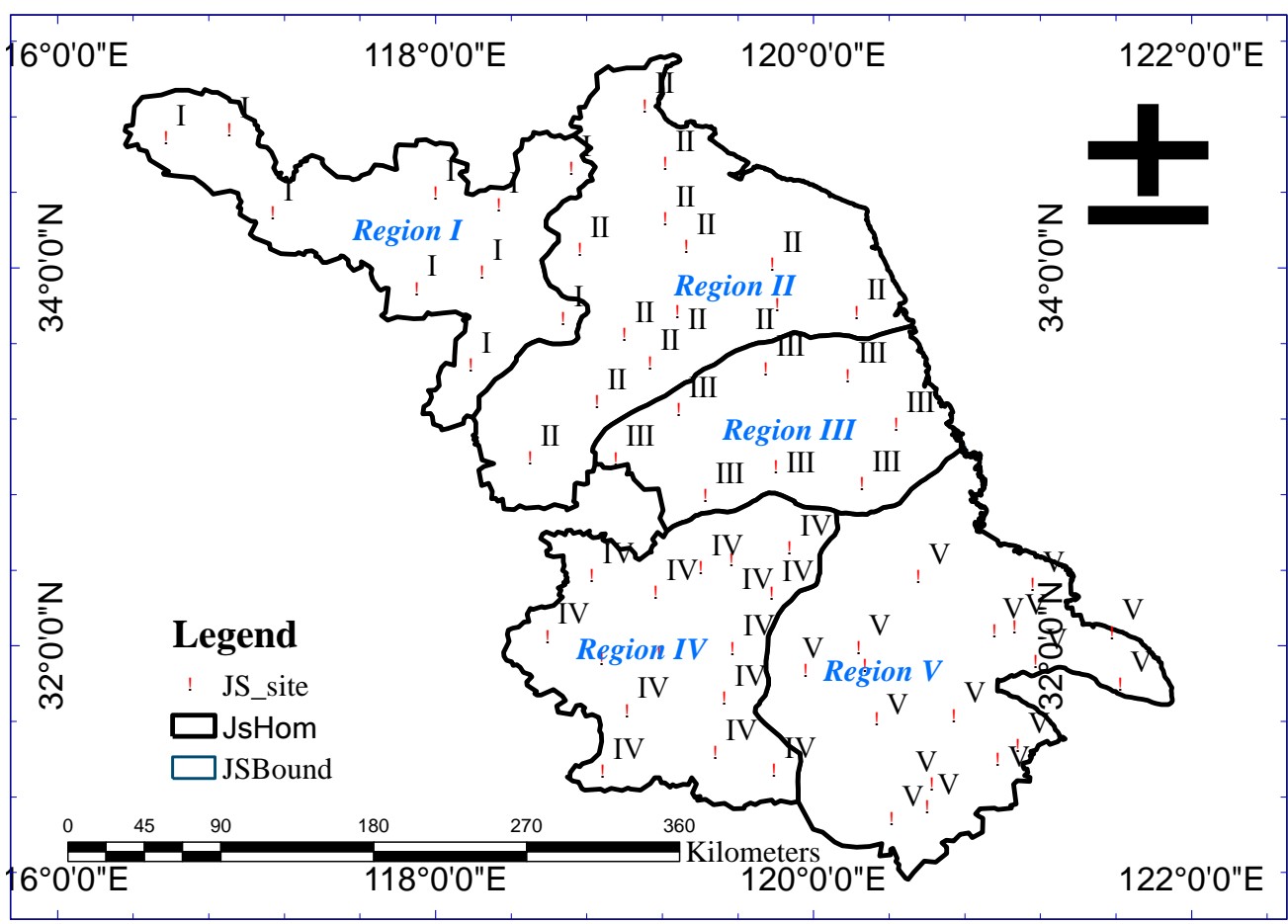

**Figure 2.** Spatial map of five homogeneous regions.

The results of the goodness-of-fit simulation experiments for the five regions are shown in Table 2. It can be seen from Table 2 that GLO and GEV are, respectively, the optimum distributions based on the two indices of |Z| and the *RMSE* for regions I and III, IV, and V. For region II, GNO is the best distribution from the |Z| value, and GEV is the best based on the *RMSE*. However, the difference between the |Z| and *RMSE* estimations is small. Therefore, GNO and GEV can be considered as the best-fitting distributions based on the two tests. However, abrupt changes in frequency estimations at the borders of adjacent homogeneous regions should be avoided. The frequency estimation has a good correlation with the tail thickness of distribution and decreases in the order of GLO, GEV, GNO, GPA, and PE3. The regions adjacent to region II have an appropriate fit with GLO. Therefore, GEV is the best distribution based on the two tests and the change in adjacent regions. Similarly, the optimum distributions of the five regions for the AES data are GNO, GPA, GPA, GPA, and GPA.

**Table 2.** Results of *MC* and *RMSE* measures for a 1-day duration.

| Test Index | Distribution | Homogeneous Region | | | | |
|:---:|:---:|:---:|:---:|:---:|:---:|:---:|
| | | **I** | **II** | **III** | **IV** | **V** |
| **Z** | GLO | −0.17 | 2.63 | −0.28 | 2.00 | 2.47 |
| | GEV | −1.39 | 0.26 | −1.66 | 0.05 | 0.37 |
| | GNO | −2.05 | −0.25 | −2.05 | −0.78 | −0.56 |
| | GPA | −4.46 | −5.11 | −4.87 | −4.71 | −4.79 |
| | PE3 | −3.21 | −1.35 | −2.83 | −2.27 | −2.24 |
| $Z_{min}$ | | GLO | GNO | GLO | GEV | GEV |
| *RMSE* | GLO | 0.0404 | 0.0650 | 0.0401 | 0.0591 | 0.0655 |
| | GEV | 0.0424 | 0.0445 | 0.0509 | 0.0395 | 0.0447 |
| | GNO | 0.0834 | 0.0466 | 0.0604 | 0.0452 | 0.0470 |
| | GPA | 0.0555 | 0.0885 | 0.1239 | 0.0811 | 0.0773 |
| | PE3 | 0.1057 | 0.0546 | 0.0797 | 0.0632 | 0.0628 |
| $RMSE_{min}$ | | GLO | GEV | GLO | GEV | GEV |

*3.2. Comparison between Exceedance Frequency and Exceedance Probability*

The regional L-moments analysis is applied to obtain the quantiles at each station on a region-by-region basis. The exceedance frequencies from 2-year to 100-year return periods are calculated station-by-station and averaged first over the region and then over the study area. The data exceedance frequencies at each station for study area are found from Table S1 in Supplementary Material. The average region-by-region exceedance frequencies are shown in Table 3 over the entire study area. It can be seen from Table 3 that the average exceedance frequencies are higher than the corresponding theoretical exceedance probabilities for 2-year to 100-year return periods over the study area, which are 0.507, 0.206, 0.111, 0.045, 0.021, and 0.011 for 2-year, 5-year, 10-year, 25-year, 50-year, and 100-year return periods, respectively. The corresponding real return periods are calculated to be 1.97 years, 4.85 years, 8.99 years, 22.25 years, 47.49 years, and 91.87 years, which indicates that extreme precipitation events occur more frequently. These data indicate that current quantile estimates based on AMS data are underestimated for frequent events in the study area.

**Table 3.** Average exceedance frequencies for the study area.

| Region | Return Period (R.P.)/Exceedance Probability (E.P.) | | | | | |
|:---:|:---:|:---:|:---:|:---:|:---:|:---:|
| | **2-yr** | **5-yr** | **10-yr** | **25-yr** | **50-yr** | **100-yr** |
| | **0.50** | **0.20** | **0.10** | **0.04** | **0.02** | **0.01** |
| I | 0.518 | 0.210 | 0.124 | 0.051 | 0.022 | 0.012 |
| II | 0.505 | 0.201 | 0.106 | 0.045 | 0.021 | 0.011 |
| III | 0.500 | 0.201 | 0.120 | 0.044 | 0.020 | 0.010 |
| IV | 0.508 | 0.217 | 0.101 | 0.041 | 0.022 | 0.013 |
| V | 0.502 | 0.202 | 0.106 | 0.044 | 0.021 | 0.009 |
| *Average E.P.* | 0.507 | 0.206 | 0.111 | 0.045 | 0.021 | 0.011 |
| *Real R.P.* | 1.97-yr | 4.85-yr | 8.99-yr | 22.25-yr | 47.49-yr | 91.87-yr |

*3.3. Verification of the Applicability of Chow's Equation in the Study*

The underestimated frequencies from the AMS data can be revised for low-return periods if Chow's equation is applicable to this study area. The procedure for verification is as follows: First, the quantiles based on real AES and AMS data are independently estimated for the study area (Table 4). Then the AES–AMS ratios are obtained based on frequency estimates from 2-year to 100-year return periods. Second, frequency estimations and their ratios are calculated based on real AMS data, where AES is obtained based on

AMS data and Chow's equation. The best-fit distribution of each homogeneous region is used to calculate the frequency estimates. The verification results are shown in Figure 3.

**Table 4.** Quantile estimates of extreme precipitation for a 1-day duration with different return periods in homogeneous region I.

| Site Name | Quantile Estimates Based on AES Data | | | | | |
|---|---|---|---|---|---|---|
| | **2-yr** | **5-yr** | **10-yr** | **25-yr** | **50-yr** | **100-yr** |
| Fengxian | 92.2 | 117.3 | 139.7 | 175.1 | 206.8 | 243.3 |
| Peixian | 98.4 | 125.2 | 149.1 | 186.9 | 220.7 | 259.7 |
| Pizhou | 106.6 | 135.6 | 161.5 | 202.4 | 239.0 | 281.2 |
| Xuzhou | 103.2 | 131.3 | 156.4 | 196.0 | 231.5 | 272.4 |
| Xinyi | 98.2 | 124.9 | 148.8 | 186.4 | 220.2 | 259.1 |
| Donghai | 98.9 | 125.9 | 149.9 | 187.9 | 221.9 | 261.1 |
| Suining | 115.4 | 146.8 | 174.9 | 219.2 | 258.9 | 304.6 |
| Suyu | 116.5 | 148.2 | 176.5 | 221.2 | 261.3 | 307.5 |
| Siyang | 104.9 | 133.4 | 158.9 | 199.1 | 235.2 | 276.7 |
| Sihong | 104.5 | 132.7 | 154.9 | 185.5 | 209.6 | 234.5 |
| | Quantile estimates based on AMS data and Chow's equation | | | | | |
| Fengxian | 91.0 | 116.6 | 139.1 | 174.9 | 207.9 | 246.9 |
| Peixian | 99.4 | 127.4 | 152.0 | 191.1 | 227.2 | 269.8 |
| Pizhou | 106.0 | 135.9 | 162.1 | 203.8 | 242.2 | 287.6 |
| Xuzhou | 103.9 | 133.2 | 158.8 | 199.7 | 237.3 | 281.9 |
| Xinyi | 96.8 | 124.1 | 148.0 | 186.1 | 221.2 | 262.7 |
| Donghai | 99.0 | 126.9 | 151.4 | 190.3 | 226.2 | 268.6 |
| Suining | 111.7 | 143.2 | 170.8 | 214.7 | 255.2 | 303.1 |
| Suyu | 111.9 | 143.4 | 171.0 | 215.0 | 255.5 | 303.5 |
| Siyang | 105.6 | 135.3 | 161.4 | 202.9 | 241.1 | 286.4 |
| Sihong | 100.0 | 128.2 | 152.9 | 192.2 | 228.5 | 271.4 |
| *Mean RE (%)* | 1.72 | 1.60 | 1.41 | 1.68 | 2.49 | 3.64 |

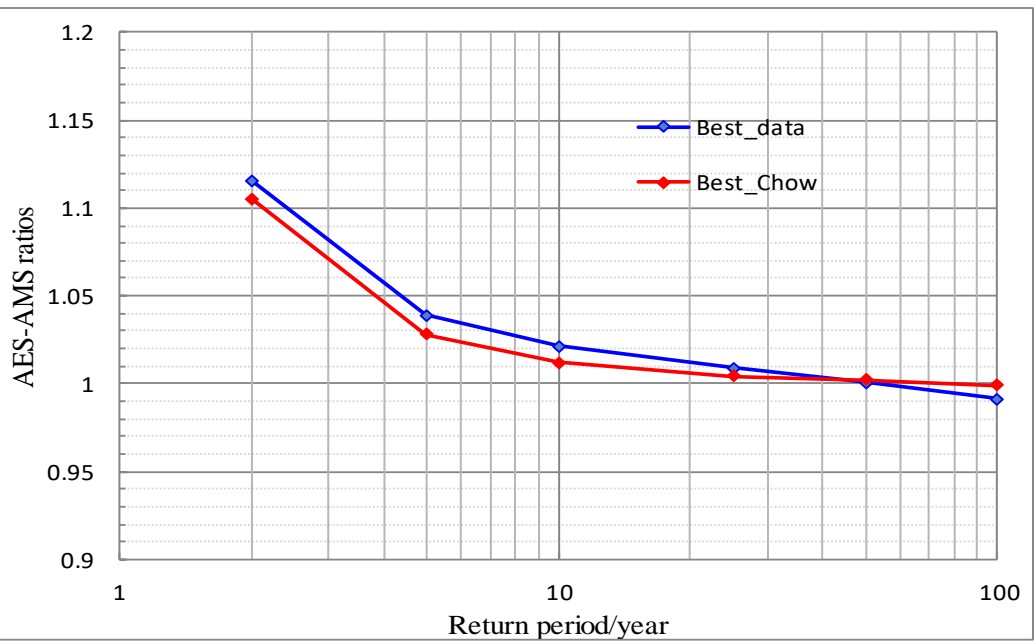

**Figure 3.** Comparison of AES–AMS ratios obtained using Chow's equation and real data.

Taking Region I as an example, quantile estimates based on AES data, AMS data, and Chow's equation are shown in Table 4. The results from region II, III, IV, and V can be found in Table S2 in Supplementary Material. The frequency estimates obtained from the

real AES data have good consistency with that obtained from computed AES data. The mean relative error from 2-year to 100-year return periods is 2.09%, indicating that the AMS data and Chow's equation may be used as an alternative method when only AMS data are available. Figure 3 indicates that the general trend is consistent between Chow's equation and the data. The AES–AMS ratio gradually decreases with increasing length of the return period; ratios are >1 when the recurrence intervals are less than 25 years, near 1 between 25-year and 50-year return periods, and <1 for return periods longer than 50 years. However, the magnitude of the decrease is largest when the recurrence interval is less than 25 years. It can also be seen from the curve trend that different sampling methods can have a substantial effect on the low-return-period interval. Taken overall, the best-fit Chow's case is consistent with the best-fit real data, indicating that Chow's equation can be used as a simple method to obtain reasonable AES-AMS ratios consistent with those obtained from real AES and AMS data.

### 3.4. Reliable Frequency Estimation and Spatiotemporal Analysis

By applying Chow's equation, quantiles derived from AMS data can be revised at low return periods. A set of rational and reliable frequency estimations can be obtained using a regional L-moments method based on the AMS data and Chow's equation. Solving the quantile for a 1-year return period is most important, which means solving the integral lower limits of the frequency distribution curve. For example, frequency estimates for different return periods in region I are shown in Table 4. It can be seen that the estimates increase incrementally with the length of the recurrence interval as a whole. The maximum estimation occurs at the Suyu station. We compared the quantile estimates with the maximum of the 24-h observation series for each station, which can indirectly reflect estimation accuracy to some extent due to the unknown true value of the frequency estimate. Considering the 24-h record length (51 years) of the Suyu station, frequency estimates for the 50-year recurrence interval were selected to assess consistency. The quantile estimate is 255.5 mm, which is consistent with the maximum observed 24-h value (253.9 mm). The frequency estimates also agree with observations at other sites and provide a scientific basis for flood disaster warnings and urban construction, among other uses.

Figure 4 shows the spatial mapping of frequency estimates for a 24-h duration for 1-year, 10-year, 25-year, and 50-year return periods, which have similar patterns. The estimated values at the northern end of the study area are greater than those at the southern area, and all of the estimates increase with increased length of the return period. The highest frequency estimates are observed near the Suqian and Lianyungang stations in the northern part of Jiangsu, and low values are observed in the southern Taihu lake basin. These data suggest that Xuzhou and Lianyungang are in a high-risk area of extreme precipitation that may be subject to flash floods. Therefore, decision makers should pay heightened attention to the risk of flooding and water resource management in these areas.

### 3.5. Validation of Frequency Estimations of Extreme Precipitation

Because the true value of the frequency estimation is unknown, the accuracy of the estimated value cannot be evaluated using the error of the estimated value and the true value. However, the accuracy of quantiles is indirectly reflected by a comparison of the estimation and observation at the same frequency. The plotting-position estimator is used to compute the experience frequency. The experience frequency is defined as follows:

$$P = (i + A)/(n + B) \tag{11}$$

where $i$ is the sequence number from ascending series, $n$ is the number of sequence length for each site, $A$ and $B$ are the parameters, $A$ is equal to $-0.35$, and $B$ is zero [13].

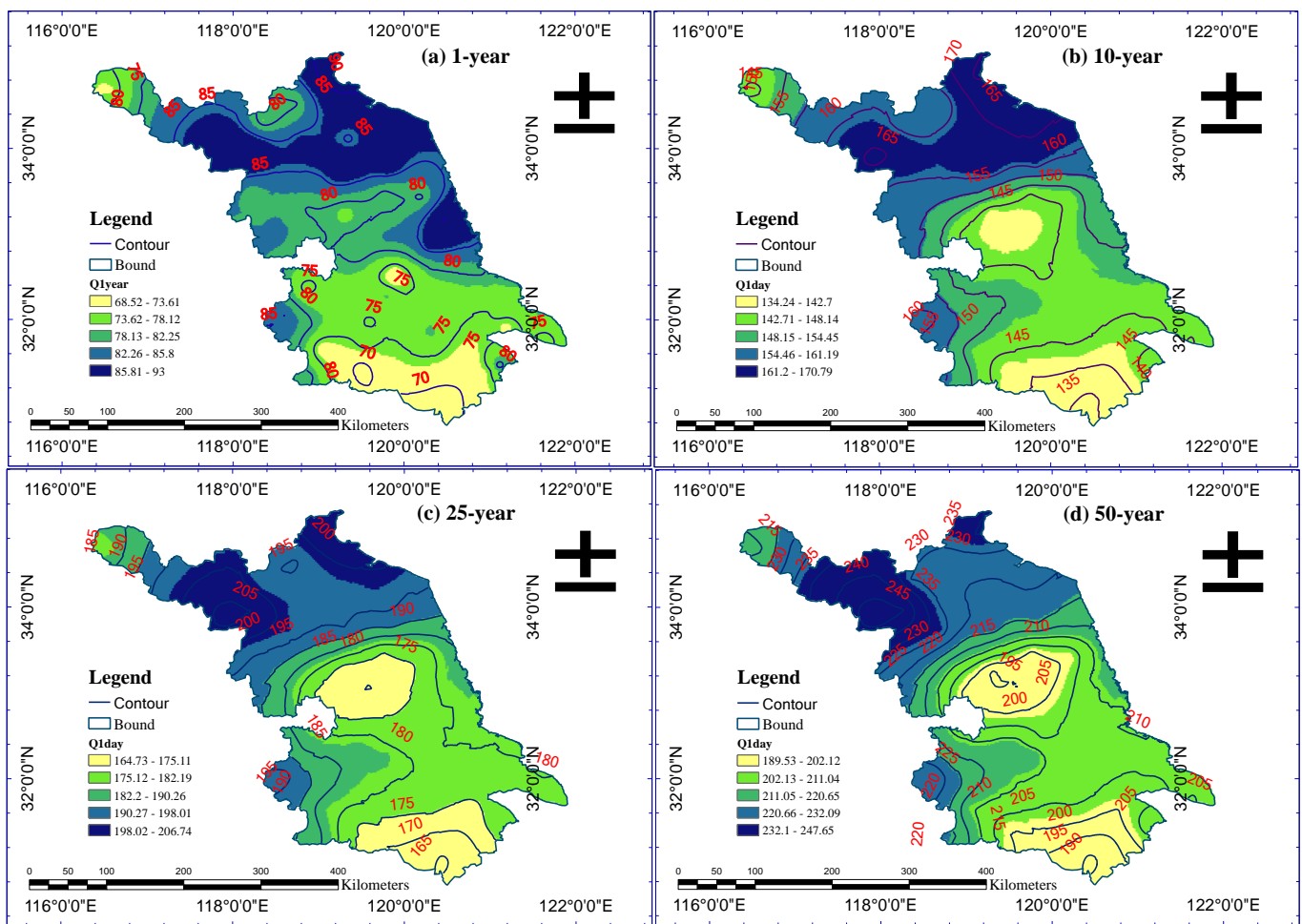

**Figure 4.** Map of quantile estimates for (**a**) 1-year, (**b**) 10-year, (**c**) 25-year, and (**d**) 50-year return periods.

Three statistical criteria, *RE*, *RMSE*, and the correlation coefficient (*r*) are used to judge the precision of frequency estimates. The results can be found in Table 5. The average *RE*, *RMSE*, and *r* of all stations in the study area are 5.56%, 0.107 mm, and 0.969, respectively. This indicates that the estimation is in good agreement with the observation. Figure 5 shows a scatterplot of measured and observed quantiles at the same frequency for each site in each homogeneous region. It can be seen from Figure 5 and Table 5 that the simulation is consistent with the set of observations as a whole, with a value of r greater than 0.96 in each homogeneous region. From the above, it may be concluded that the frequency estimation is reasonable and reliable for low return periods. As a whole, frequency estimations based on the regional L-moments method are in good agreement with observations.

**Table 5.** Comparisons between estimation and observation.

| Homogeneous Regions | *RE* (%) | *RMSE* (mm) | *r* |
|---|---|---|---|
| Region I | 5.47 | 0.101 | 0.975 |
| Region II | 5.31 | 0.093 | 0.975 |
| Region III | 4.77 | 0.09 | 0.971 |
| Region IV | 5.88 | 0.118 | 0.965 |
| Region V | 5.96 | 0.119 | 0.961 |
| All | 5.56 | 0.107 | 0.969 |

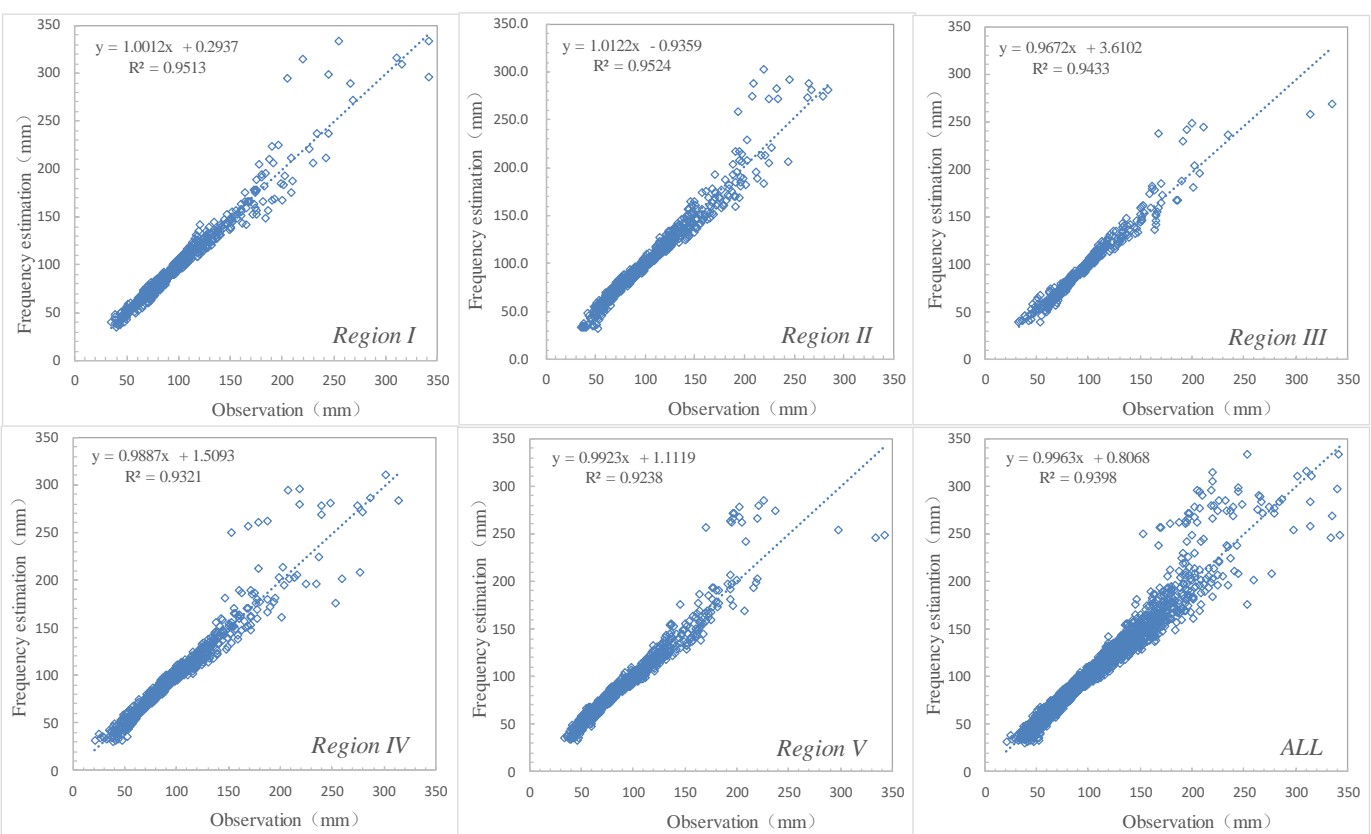

**Figure 5.** Scatterplot of estimations and observations at the same frequency for each homogeneous region.

## 4. Discussion

The results of this study are valuable for revising the underestimation of quantiles and obtaining a set of reliable quantile estimates in the study area. However, some issues may benefit from a more in-depth analysis in future research.

The determination of homogeneous region is an important step in regional frequency analysis. The optimum distributions are, respectively, GEV and GPA distribution based on the AMS data and AES data for most regions of the study area, which is consistent with many previous studies. Many research studies have shown that GEV is the most commonly used for AMS analysis, and GPA is frequently proposed for PDS analysis [35–39]. However, the national guidelines and regulation for calculation design storm and flood in China recommend the use of PE3 distribution, which is inconsistent with the research in this paper [40]. The main attribution includes that PE3 is a recommended choice based on the conventional moment method, which is very different from when sample size is small, or the skewness of the sample is considerable. Many research studies have proved that L-moments are less subject to bias in estimation and enable more reliable inferences to be made from small samples than conventional moment method [22,41–43]. Therefore, the optimum distribution of this paper is rational and reliable based on the summary of judgement. At the same time, identification of homogeneous region may cause the discontinuity around the boundary of adjacent regions. Very few papers dealing with the discontinuity are found in the literature. So, spatial consistency should be considered and further research in the next study.

Research has shown that the quantiles based on AMS data are underestimated at low return periods in this paper, which is in accordance with previous research and theories [1,18,20,22]. Lin et al. verified the result that exists a significant underestimation based on 1438 stations data in southwestern United States [1,20]. Frequency estimates are underestimated but the magnitude of underestimation is not obvious in this study. Some possible reasons are analyzed and discussed as follows. By analyzing the AMS data of

the site, it is found that a negative correlation exists between the exceedance frequency and skewness coefficient of the station; that is, the larger the positive skewness coefficient, the smaller the exceedance frequency. The frequency distribution diagram based on the AMS data is used to analyze the causes (Figure 6). Taking Region I and Region V for example, the stations with the largest L-Cs (58013 and 58,349 sites) and with the smallest L-Cs (58,131 and 58,345 sites) are selected to analyze the underestimated reasons in this study. It can be seen from Figure 6 that the station with the largest L-Cs have the maximum rainfall value in the corresponding region, and the AMS sequence with the smallest L-Cs is approximate to normal distribution, which has uniform and continuous characteristics and no extra-large value. We may come to the conclusion that the distribution of sparse and discontinuous extreme precipitation data at large value intervals is the main factor resulting in low exceedance frequency values. Second, factors including small sample sizes and data series of inadequate length can also affect the calculation of the exceedance frequency. Therefore, a larger range and longer sequence of data should be collected and analyzed in future research.

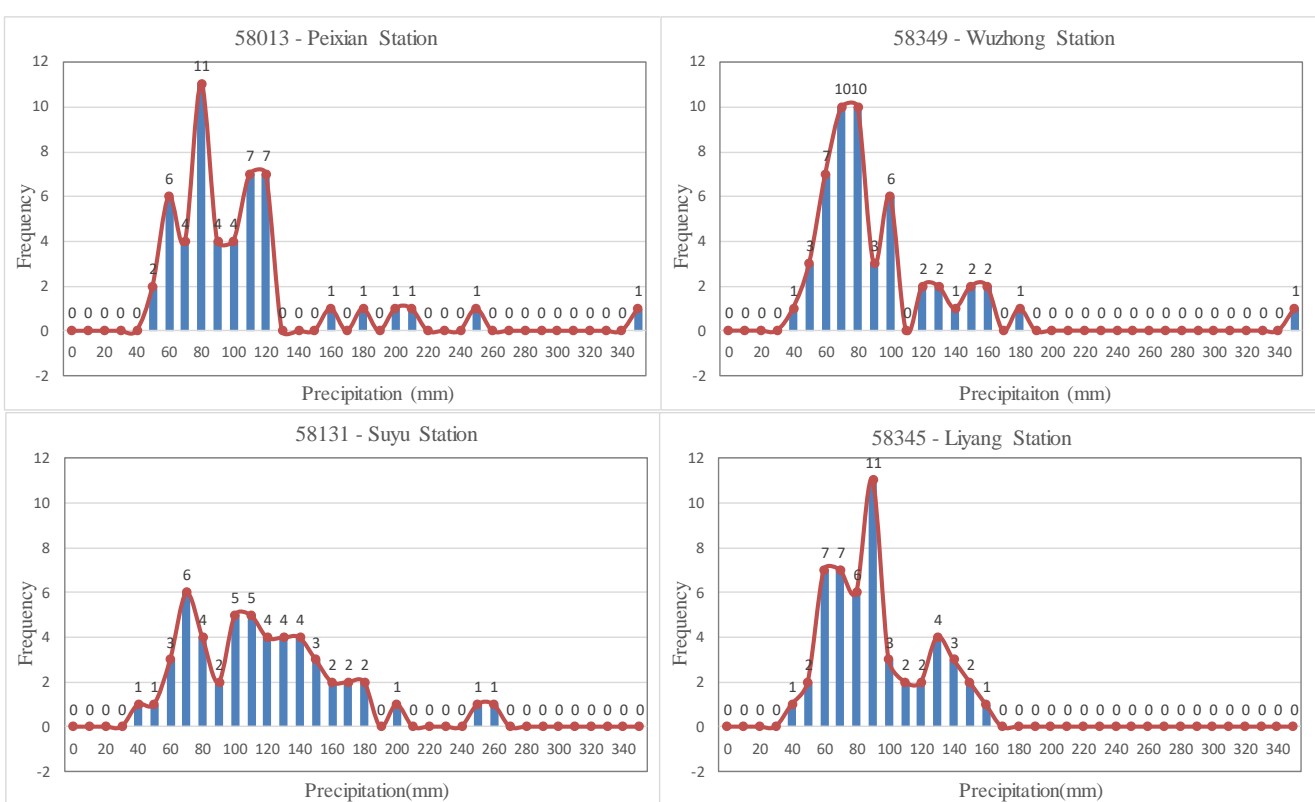

**Figure 6.** Frequency diagram of AMS data at representative site.

A set of rational and reliable frequency estimations can be obtained based on the abovementioned two methods, especially quantiles for the 1-year return period in this study. It is the main innovation of this paper that was missing from previous works. Many researchers have solved the quantile estimates for greater than or equal to a 2-year recurrence interval in regional frequency analysis [16,44–46]. Only few papers solving the quantiles of a 1-year return period are found in the literature [47–49]. However, these studies assume that the non-exceedance probability of a 1-year return period is equal to 0.1 because it is not computable if the non-exceedance probability equals zero. The assumption has no theoretical basis, and the quantile results are inaccurate at lower recurrence intervals due to the underestimation of the AMS analysis. On the basis of verification of Chow's equation applicability, the quantiles can be corrected based on AMS data at low-return periods. It is a very valuable practical element in China because only

AMS data is available from China Meteorological Administration since the 1960s [50]. As a whole, frequency estimations based on the regional L-moments method are in good agreement with observations. The findings validate the reliability of frequency estimations at low return periods. However, the quantiles are slightly overestimated or underestimated in cases of large extreme precipitation. Potential reasons for the deviation of quantiles at large return periods include spatial inconsistencies around the boundary of adjacent regions, short sampling series, and other factors. Future research should conduct a more in-depth analysis of these possibilities.

It is a pity that hourly extreme precipitation data were not available for this study. Ideally, complete quantile estimates from 1-h to 30-day durations would be carried out in the future in the study region. Thus, a complete set of spatiotemporal frequency estimates from multi-duration and multi-return periods can be obtained in the region, which can provide more of a quantitative and scientific basis for decision making. Such data would provide a reference criterion of different duration for comparison, and also provide a stronger scientific basis for issuing storm disaster and flash flood warnings.

## 5. Conclusions

In this paper, a regional frequency analysis of extreme precipitation in the province of Jiangsu in the Yangtze River Delta was studied using regional L-moments methods. A set of rational and reliable frequency estimation was obtained based on AMS data and Chow's equation. Some of the main findings obtained from the research are as follows:

The study area is categorized into five homogeneous regions using cluster analysis. Five distributions (GLO, GEV, GNO, GPA, and PE3) are investigated, and MC simulations and RMSE tests are used to identify the optimum distribution in each homogeneous region. The best-fit distributions based on AMS data are GLO, GEV, GLO, GEV, and GEV for the five homogeneous regions, respectively. The best-fit distributions based on AES data are GNO, GPA, GPA, GPA, and GPA, respectively. By comparing exceedance frequencies with exceedance probabilities it can be seen that extreme precipitation events occur more frequently, and that current quantile estimates based on AMS data are underestimated for frequent events in the study area.

Verification of Chow's equation in this study area shows that there is generally good consistency between real AES data, and AES data generated using Chow's equation and real AMS data. As a whole, the results indicate that Chow's equation can be used as a simple method to obtain reasonable AES-AMS ratios, similar to those obtained from real AES and AMS data. This finding also means that frequency estimations can be revised at lower return periods based on real AMS data and Chow's equation. Two methods can be used to correct for underestimation of frequency estimates. The first method is to use AES data in combination with theoretical exceedance probabilities, such as 0.5, 0.2, 0.1, 0.04, and 0.02 for the corresponding return periods of 2 years, 5 years, 10 years, 25 years, and 50 years. The second way is to use AMS data in combination with the correction of return periods based on Chow's conversion equation. The two methods are equivalent in quantile estimation. However, the second method is strongly recommended due to its simple data processing requirements and reliable results, especially when only AMS data are available for the study area.

A set of rational and reliable frequency estimations can be obtained using the regional L-moments method based on AMS data and Chow's equation. Solving the quantile for a 1-year return period is most important, which means the integral lower limits of the frequency distribution curve. The results show that the estimates increase incrementally with the recurrence interval, and that the estimates agree with observations as a whole. The spatial mapping of quantiles shows that similar patterns exist for 1-year, 10-year, 25-year, and 50-year return periods, and that quantiles in the northern part of the study are greater than in the southern area. The highest frequency estimates are observed near the Suqian and Lianyungang stations in the northern part of Jiangsu, and low values are observed in the southern Taihu lake basin. This suggests that the Xuzhou and Lianyungang areas are

likely at high risk of flash floods due to extreme precipitation. Decision makers should pay heightened attention to flood risk and water resource management in these areas.

Based on the three criteria of RE, RMSE, and r, the accuracy of estimations can be evaluated by comparing estimations and observations at the same frequency. The results show that frequency estimations are in good agreement with observations, with the average RE, RMSE, and r of all stations being 5.56%, 0.107 mm, and 0.969, respectively, especially a r > 0.96 was found in each homogeneous region. Frequency estimations based on the regional L-moments method are in good agreement with observations. The findings validate the reliability of frequency estimations at low return periods. A set of reliable quantile estimates are obtained based on two revised ways, which provide a new perspective in regional frequency analysis.

**Supplementary Materials:** The following are available online at https://www.mdpi.com/article/10.3390/w13131832/s1, Table S1: Exceedance frequencies of stations based on AMS data at each station for each region. Table S2: Quantile estimates of extreme precipitation for a 1-day duration from 2-yr to 100-yr return periods in homogeneous region II, III, IV and V.

**Author Contributions:** For research articles with several authors, Conceptualization, Y.S. and J.Z.; methodology, Y.S.; validation, Y.S., J.Z. and J.X.; formal analysis, A.F.; investigation, J.W.; data curation, J.Z.; writing—original draft preparation, Y.S.; writing—review and editing. All authors have read and agreed to the published version of the manuscript.

**Funding:** This work is financially supported by the Meteorological Open Research Fund in Huaihe River Basin (HRM201702), by the Special Fund for Natural Science Foundation of Jiangsu province (BK20141001), by the China Postdoctoral Science Foundation ( No. 2020T130309, No. 2019M651892), by the Jiangsu Water Resources Science and Technology Project (No. 2020022).

**Informed Consent Statement:** It is not applicable for studies not involving humans.

**Data Availability Statement:** The data used to support the findings of this study are available from the corresponding author upon request.

**Acknowledgments:** The authors are also grateful to Lin Bingzhang for providing some constructive suggestions. The authors would like to acknowledge the anonymous reviewers for their thoughtful comments and suggestions.

**Conflicts of Interest:** The authors declare that they have no conflict of interest.

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
