# Peer review of "Revision of Frequency Estimates of Extreme Precipitation Based on the Annual Maximum Series in the Jiangsu Province in China"

_water, doi:10.3390/w13131832_

Round 1

Reviewer 1 Report

Summary

The paper presents a precipitation extreme analysis at regional scale using a method named L-moments and Chow’s equation for a basin in China.

Authors demonstrate as frequencies of extremes are not well estimated and propose this technique for return period analysis.

General Comment.

The paper is interesting and the applied method is consistent with the main aim of the article.

Some parts should be better addressed, in particular, a better description in the methods section, which is quite confusing for those who are not familiar with this technique. Despite this consideration, analysis is well structured and conclusions supported by data.

A general English editing is required

Abstract

Line 17: sentences like “are in good agreement” should be avoided in scientific papers. It is preferable to write numbers of statistical scores, without any personal consideration.

Lines 27-29: numbers

Line 30: it is not clear what does “theatrical” means in this context

Lines 40-41: Acronyms GEV and LP3 should be explained.

Lines 65-66: authors state that less research has been carried out in China for quantiles underestimation, however, even if less than other countries, it should be useful to add one or more references for China.

Methods

Line 124: it is not clear what authors mean with the word “titled”, related to the topography. Also, it is written that elevation is “high” in northern and southern of the basin, however, from the legend in Fig. 1 I can infer that maximum elevation is 607 m a.s.l., which is not so high (those altitudes are classified as hills, not mountains, but maybe anything is relative).

Line 126: it is not fully clear what authors mean for “large” density

Line 132: maybe it is “located” and not “locates”. Also, “are (located” is missing.

Figure1: the figure is quite clear, however, I suggest t choose another color for the JSRiver, as the yellow is confused with the DEM shade.

Lines 183-184: reference for each distribution (or at least one reference where characteristics of each distribution are summarized) should be given.

Formula 3: coefficients B and sigma seems to be not described in the text.

Formula 4: this is inserted as an image, therefore, it is grainy and not readable when the article is printed

Results

Paragraph 3.1: I guess that this section should be moved to the methods, as it describes the criteria of geographical classification of the target areas.

Moreover, it would be interesting to know if this classification actually is actually corresponding to meteorological (or hydrological) warning areas that should be defined by civil protection authorities for risk assessment.

Figure: the DEM legend can be deleted, I guess, as it is not

Line 238: it is not essential to repeat twice GLO and GEV, as they are calculated for both Z and RMSE.

Line 239: the plural form for “index” is “indices”

Table 3: it is not essential to the discussion, however, for completeness, I think that same results for regions II, III, IV and V cab ne added in supplementary materials.

Line 263: it is not clear where exceedance probabilities are reported in table 4

Table 5: same comment as Table 3

Author Response

Answer to the reviewer's recommendation

Reviewer's comments:

General Comment.

The paper is interesting and the applied method is consistent with the main aim of the article. Some parts should be better addressed, in particular, a better description in the methods section, which is quite confusing for those who are not familiar with this technique. Despite this consideration, analysis is well structured and conclusions supported by data. A general English editing is required.

Answer: Authors are in agreement with the reviewer’s suggestion, and have already supplemented the related content in this manuscript.

 (1) Reviewer’s advises are good. Authors agree with the reviewer’s recommendations that the methods section needs to have a better description in this manuscript. We have already supplemented related content in this section in order to make the reader better understand the method introduction.

(2) we are very sorry for our incorrect writing in the text. Authors have already carefully examined and corrected the mistakes in grammar and writing in order to make the document readable in this manuscript.

Specific comments:

Abstract

Line 10: Invisible texts covered by Fig.1

Answer: Authors don’t find invisible texts in Line 10.

Line 17: sentences like “are in good agreement” should be avoided in scientific papers. It is preferable to write numbers of statistical scores, without any personal consideration.

Answer: The reviewer’s advice is good. Authors has revised the sentences for avoiding any personal consideration in the manuscript.

Lines 27-29: numbers

Answer: Authors don’t understand what dose “numbers” means in this context.

Line 30: it is not clear what does “theatrical” means in this context

Answer: We are very sorry for the spelling mistake. “theatrical” should be modified as “theoretical”.  Authors have already revised it in the context.

Lines 40-41: Acronyms GEV and LP3 should be explained.

Answer: Yes. Authors have already supplemented the full name of abbreviation GEV and LP3 in the manuscript.

Lines 65-66: authors state that less research has been carried out in China for quantiles underestimation, however, even if less than other countries, it should be useful to add one or more references for China.

Answer: The reviewer’s advice is good. We have already added the related references in manuscript.

Methods

Line 124: it is not clear what authors mean with the word “titled”, related to the topography. Also, it is written that elevation is “high” in northern and southern of the basin, however, from the legend in Fig. 1 I can infer that maximum elevation is 607 m a.s.l., which is not so high (those altitudes are classified as hills, not mountains, but maybe anything is relative).

Answer: The reviewer’s comments are reasonable to some extent. the word “titled” maybe express the what authors mean. We have already revised the word. Since the highest elevation is 607m, authors used “low mountains and hills”. Considering that the reviewer’s suggestion is more reasonable, we have modified this sentence in the text.

Line 126: it is not fully clear what authors mean for “large” density

Answer: Yes. The logic of this sentence is not clear. We have already revised it in the manuscript.

Line 132: maybe it is “located” and not “locates”. Also, “are (located” is missing.

Answer: Sorry for the grammatical error. It is modified in the manuscript.

Figure1: the figure is quite clear, however, I suggest t choose another color for the JSRiver, as the yellow is confused with the DEM shade.

Answer: We agree with the reviewer’s suggestion that the same color tends to confuse the reader. We have already chosen blue color for the JSRiver.

Lines 183-184: reference for each distribution (or at least one reference where characteristics of each distribution are summarized) should be given.

Answer: Yes. We have already added the related references in manuscript.

Formula 3: coefficients B and sigma seems to be not described in the text.

Answer: Thanks for the reviewer’s suggestion. The authors have already supplemented the bias the bias (B4) and standard deviation (σ 4) of the regional average L-kurtosis in the text.

Formula 4: this is inserted as an image, therefore, it is grainy and not readable when the article is printed

Answer: The reviewer’s advice is right. Authors have already edited the formula 4 and make it readable when the article is printed.

Results

Paragraph 3.1: I guess that this section should be moved to the methods, as it describes the criteria of geographical classification of the target areas.

Moreover, it would be interesting to know if this classification actually is actually corresponding to meteorological (or hydrological) warning areas that should be defined by civil protection authorities for risk assessment.

Answer: Authors agree with the reviewer’s suggestion. We have already moved this section to methods, and supplemented the related content in the manuscript. Identification of homogeneous regions is to reduce the uncertainties that exist in at-site, and to improve the accuracy of frequency estimation. Some previous researches shown that the area of the high frequency estimates of extreme precipitation is consistent with the risk areas defined by hydrometeorological department.

Figure: the DEM legend can be deleted, I guess, as it is not

Answer: Yes. The authors have already deleted the DEM legend according to the reviewer’s advice.

Line 238: it is not essential to repeat twice GLO and GEV, as they are calculated for both Z and RMSE.

Answer: Authors may not have made the sentence clear. It is not repeated twice. It just happens that the optimal distribution in these regions is the same. Authors have already modified the expression of the sentence in the manuscript.

Line 239: the plural form for “index” is “indices”

Answer: the “index” is modified as “indices” in the text.

Table 3: it is not essential to the discussion, however, for completeness, I think that same results for regions II, III, IV and V cab ne added in supplementary materials.

Answer: The reviewer’s advice is good. Authors have already moved the table 3 to the supplementary material, and added the results from region II, III, IV and V.

Line 263: it is not clear where exceedance probabilities are reported in table 4

Answer: Authors have already explained and supplemented the exceedance probabilities in Table 4.

Table 5: same comment as Table 3

Answer: The reviewer’s advice is good. However, authors think that it is essential to the discussion in the text. There are mainly two reasons as follows. First, Table 5 and Figure 3 verify the applicability of Chow’s equation from different perspectives. Figure 3 focuses on the consistency of the general trend between Chow’s equation and the data. Table 5 highlights the specific frequency estimates and their relative error at each station. Second, Table 5 is also used to verify the reliable frequency estimation and spatiotemporal analysis.

    Take Region I as an example, the frequency estimates are used to analyze and discussion in the manuscript, and the results from region II, III, IV and V are added to the supplementary material.

Reviewer 2 Report

Thank you for the opportunity to review this nicely presented piece of interesting and timely research.

The manuscript is concerned with "Revision of frequency estimates of extreme precipitation based on the annual maximum series in the Jiangsu Province in China” which is very interesting and actual in the time of climate change and high risk of flash floods due to extreme precipitation, that could be with the aim and scope of the journal.

The paper is technical sound in its present form and could be publicate in the Water journal.

  • Abstract & introduction: are focused on the main aim of the paper and the new contribution of authors to the state of art. Improving the previous works by verifying underestimation of the quantiles and the provision of two new methods for obtaining reliable quantile estimations of extreme precipitation at lower recurrence intervals, especially in solving reliable estimates for a 1-year return period from the integral lower limit of the frequency distribution could be a very good basis for flood control and water resource management not only for this chosen part of the world.
  • Materials & methods: Used methods seems adequate and obtained results are very well presented. The authors combined different research methods (Regional L-moments method, index flood procedure, Monte Carlo simulation, Root mean square error, AMS, SAS) for the area of Jiangsu province in China. The suitability and technical standards of the methods are described with sufficient details of the processes so that another researcher is able to reproduce the methods and calculation for different similar areas over the world.
  • Results & discussion: The data are well controlled and robust and authors provided relevant and current references. Conclusions are strictly based on actual facts and figures. In my opinion they carried out sufficient and appropriate statistical analyses.
  • Conclusion: Authors provided adequate proof for their claims. Also they know and wrote about negatives and future direction of their research (small sample sizes and data series of inadequate length, only AMS data available)  and how it can be done in the future (a larger range and longer sequence of data; spatial consistency). Their findings provide useful guidance in the areas at high risk of  flash floods due to extreme precipitation. The results show that quantile estimations are in good agreement with observations, thereby providing a robust basis for flood control and water resource management. 
  • All the references cited are relevant and adequate.

Author Response

Authors thanks for the reviewer’s comments and recognition concerning our manuscript entitled “Revision of frequency estimates of extreme precipitation based on the annual maximum series in the Jiangsu Province in China”(ID: WATER-1253116). The reviewers’ comments are very helpful for improving our paper, as well as the important guiding significance to this research. According to the deficiencies found in the current research, authors will do in-depth analysis and study in the future, such as a larger range, longer sequence data, spatial consistency and so on. Thanks for reviewer’s suggestion again.